



# Identification of Hotspots of Rainfall Variation
# Sensitive to Indian Ocean Dipole Mode through
# Intentional Statistical Simulations
Jong-Suk Kim[1], Phetlamphanh Xaiyaseng[1], Lihua Xiong[1], Sun-Kwon Yoon[2,*], Taesam Lee[3,*]
[1] State Key Laboratory of Water Resources and Hydropower Engineering Science, Wuhan
University, Wuhan, 430072, P.R. China; jongsuk@whu.edu.cn (J.K.); lar99@yahoo.com
(P.X.); xionglh@whu.edu.cn (L.X.)
[2] Department of Safety and Disaster Prevention Research, Seoul Institute of Technology, Seoul
03909, Republic of Korea
[3] Department of Civil Engineering, ERI, Gyeongsang National University, 501 Jinju-daero,
Jinju, Gyeongnam, South Korea, 660-701
[*] Correspondence: skyoon@sit.re.kr (S.Y.); tae3lee@gnu.ac.kr (T.L.)
**Abstract.** This study analyzed the sensitivity of rainfall patterns over the Indochina
Peninsula (ICP) to sea surface temperature in the Indian Ocean based on statistical
simulations of observational data. Quantitative changes in rainfall patterns over the ICP
were examined for both wet and dry seasons to identify hotspots sensitive to ocean
warming in the Indo-Pacific sector. Rainfall variability across the ICP was confirmed
amplified by combined and/or independent effects of the El Niño–Southern Oscillation
and the Indian Ocean Dipole (IOD). During the years of El Niño and a positive phase of
the IOD, rainfall is less than usual in Thailand, Cambodia, southern Laos, and Vietnam.
Conversely, during the years of La Niña and a negative phase of the IOD, rainfall
throughout the ICP is above normal, except in parts of central Laos and northern
Vietnam. This study also simulated the change of ICP rainfall in the wet and dry
seasons according to intentional IOD changes, and IOD-sensitive hotspots were





verified through quantitative analysis. The results of this study provide clear
understanding both of the sensitivity of regional precipitation to the IOD and of the
potential future impact of statistical changes regarding the IOD in terms of
understanding regional impacts associated with precipitation in a changing climate.
***Keywords***: Rainfall variability, Indian Ocean Dipole, ENSO, IBB simulation
**1. Introduction**
Spatiotemporal variation in precipitation extremes can result from amplification of
changes in atmosphere–ocean interactions and intensification of the hydrological cycle
on both regional and global scales attributable to the effects of global climate change
(Allan and Soden, 2008; Kim and Jain, 2011; Ge et al., 2017; Kang et al., 2017; Kim et
al., 2017; Gao et al., 2019). Changes in the magnitude and frequency of regional rainfall
are related closely to the occurrence of floods and droughts. They have important
implications not only in terms of their socioeconomic impact, but also in relation to the
management of local and/or regional hydropower, irrigation, and environmental water
resources (Chi et al., 2016; Gu et al., 2017; Choi et al., 2018). The occurrence of
extreme precipitation, which is highly likely to continue into the future, is increasingly
regarded as an area of concern by the public because many countries have experienced
such extreme events in recent years (Croitoru et al., 2013; IPCC, 2013; Hirsch and
Archfield, 2015; Chi et al., 2016; Donat et al., 2016). In particular, there has been rapid
increase in both the amount of damage and the number of fatalities associated with the
occurrence of extreme rainfall in developing countries because of their vulnerable
infrastructure, high density of human activities, and poor practices of land use and
development (Mirza, 2003; Yin et al., 2011).





The El Niño–Southern Oscillation (ENSO) is known for its active and predictable
short-term behavior within the global climate system (Chen and Cane, 2008),
characterized by irregular but periodic change in the behavior of winds and sea level
temperatures over the tropical eastern Pacific Ocean. Since the 2000s, new forms of El
Niño have appeared more frequently in the central Pacific (Ashock and Yamagata,
2009; Pradhan et al., 2011). However, little is yet known about the causes of these new
types of El Niño, some of which have been reported to have noticeable effect on the
supply of warm seasonal freshwater and hydrological extremes in Pacific Rim
countries (Kim et al., 2012; Yoon et al., 2013; Son et al., 2014; Wang et al., 2014; Kim
et al., 2017). Research over the past two decades has identified a distinct climate
anomaly in the Indian Ocean, known as the Indian Ocean Dipole (IOD) (Piechota et al.,
1998; Saji et al., 1999; Mahala et al., 2015; Lqbal and Hassan, 2018). The IOD is an
atmosphere–ocean coupling mode characterized by the opposition of anomalies of sea
surface temperature (SST) in the west and east of the tropical Indian Ocean (Piechota et
al., 1998; Saji et al., 1999; Webster et al., 1999). A positive (negative) IOD pattern is
characterized by water warmer (cooler) than normal in the western tropical Indian
Ocean (10° S–10° N, 50°–70° E) and water cooler (warmer) than normal in the
southeastern tropical Indian Ocean (10° S to the equator, 90°–110° E). These events
usually begin around May or June and they terminate rapidly in early winter after
reaching a peak between August and October (Saji et al., 1999). Long-term climatic
change has high correlation with large-scale atmospheric teleconnections and it has
been reported predictable in relation to the behavior of nonlinear climate systems,
particularly in terms of ocean-related climatic drivers such as ENSO and the IOD mode
(Piechota et al., 1998; Saji et al., 1999). ENSO and IOD patterns are known as leading
causes of large atmospheric change and they are related closely to seasonal variations in





precipitation in the Indian Ocean region and around the world (Ashok et al., 2001;
Ashok et al., 2003; McFadden et al., 2006; Pradhan et al., 2011).

Recent studies have suggested that the observed slowdown in the rise of global

mean surface atmospheric temperature is related closely to the considerable transport of
heat from the Pacific Ocean into the Indian Ocean via the Indonesian Throughflow
(Kosaka and Xie, 2013; Lee et al., 2015; Liu et al., 2016; Zhang et al., 2018).
Investigation of Indo-Pacific thermocouples can help both to improve understanding of
regional-scale climatic variability that is globally relevant and to diagnose
quantitatively such variability in a changing climate (Zhang et al., 2018). However,
there has been little previous quantitative research on rainfall variation across the
Indochina Peninsula (ICP) in relation to IOD phenomena and ENSO evolution.
Therefore, based on historical observations, this study undertook quantitative analysis
of the changes in SST in the Indo-Pacific sector and the associated interseasonal
variation of precipitation over the ICP. The study had three primary areas of interest: (1)
the spatiotemporal changes in magnitude and frequency of precipitation during the dry
and wet seasons, (2) the relationship between the changes in weather extremes and
large-scale climatic patterns over the ICP, and (3) identification of IOD-sensitive
hotspots using the intentionally biased bootstrapping (IBB) technique based on limited
historical observations.

**2. Data and Methods**
**2.1. Precipitation Dataset and Climate Change Indices**
This study used the high-resolution (0.5° × 0.5°) daily Climate Prediction Center
Global Unified Precipitation dataset for 1979–2018, which was obtained from the
website of NOAA's Earth System Research Laboratory's Physical Research Division





(https://www.esrl.noaa.gov/psd/). The Global Precipitation Climatology Center
monthly precipitation dataset with $1.0° \times 1.0°$ spatial resolution for the period 1948–
2018, which is based on quality-controlled data from 67,200 stations worldwide
(Schneider et al., 2016), was also used to identify seasonal precipitation variability
over the ICP region (5°–25° N, 90°–115° E) (Fig. 1). To identify changes in the
frequency and intensity of rainfall, six major climate change indices (Karl et al., 1999)
based on the daily Climate Prediction Center data from 1979–2018 were analyzed for
both the wet season (May–October) and the dry season (November–April). These
indices included the seasonal total precipitation (PRCPTOT) on wet days, seasonal
total of the 95th percentile of precipitation (R95pTOT) on wet days (≥1.0 mm),
seasonal maximum 1-day precipitation (RX1day), simple precipitation intensity index
(SDII) with a daily precipitation amount on wet days of ≥1.0 mm, maximum number of
consecutive dry days (CDD) with a daily precipitation amount of <1.0 mm, and
maximum number of consecutive wet days (CWD) with a daily precipitation amount of
≥1.0 mm.

**2.2. Indian Ocean Dipole (IOD) and El Niño–Southern Oscillation (ENSO)**
The monthly SST anomaly (SSTA) from NOAA's Extended Reconstructed Sea
Surface Temperature (ERSST) dataset v5 in the Tropical Indian Ocean (TIO) was
used to calculate the IOD mode index. This is defined as the SSTA difference
between the western (10° S–10° N, 50°–70° E) and southeastern (10° S to the equator,
90°–110° E) regions of the TIO (Saji et al., 1999). From 1948–2017, a 3-month running
average was applied to the IOD mode index data (August–September–October), which
is the peak phase period, with ±1σ to determine the years with positive and negative
modes of the IOD (Fig. 2). To characterize different types of ENSO event, monthly





Niño3 (5° S–5° N, 150° E–90° W) and Niño4 (5° S–5° N, 160° E–150° W) data for the
period 1948–2018 were used for El Niño development phases (December–January–
February). In this study, the pattern of El Niño was divided into two groups depending
on where the peak and persistent anomalies in SST occurred in the tropical Pacific: (1)
Eastern Pacific (EP); El Niño occurring in the EP and (2) Central Pacific (CP); El Niño
emerging in the CP. This study employed two new indices (Eq. 1) to identify the two
types of El Niño event through a simple transformation of the Niño3 and Niño4 indices,
as proposed by Ren and Jin (2011):

$$N_{CT} = N_3 - \alpha N_4$$

$$\alpha = \begin{cases} 0.4, N_3 N_4 > 0 \\ 0, otherwise. \end{cases} \quad (1)$$

$$N_{WP} = N_4 - \alpha N_3,$$

Here, $N_3$ and $N_4$ indicate the Niño3 and Niño4 indices, respectively.
Assessment of the relative impacts of the IOD and ENSO on rainfall across the ICP
was based mainly on composite analyses. During 1979–2018, the effects of ENSO and
the IOD were evaluated in terms of rainfall across the ICP during both the wet season
(May–October) and the dry season (November–April).

**2.3. Trend Detection**

A nonparametric Mann–Kendall test is commonly used to detect a monotonic pattern in
a time series of climate data based on the null hypothesis that the data are independent
and sorted randomly (Mann, 1945; Kendall, 1990). The null hypothesis $H_0$ is random in
the order of the sample data ($X_i, i = 1, 2..., n$) and it has no trend, whereas the alternative
hypothesis $H_1$ represents the monotonous tendency of $X$. The $S$ statistic for Kendall's
tau is calculated as follows:





$$S = \sum_{i=1}^{n-1} \sum_{j=i+1}^{n} sgn(X_j\text{-}X_i)$$


(2)

and
$$sgn(\_) = \begin{cases} 1 & \text{if } \_ > 0 \\ 0 & \text{if } \_ = 0 \\ -1 & \text{if } \_ < 0 \end{cases}.$$
(3)

The $S$ statistic is calculated using the following mean and variance:
$$E(S) = 0,$$
(4)

$$V(S) = \frac{n(n-1)(2n+5)\text{-}\sum_{m=1}^{n} t_m m(m-1)(2m+5)}{18},$$
(5)

where $t_m$ measures the ties of extent $m$. The standardized test statistic $Z$ is estimated as
follows:
$$Z = \begin{cases} \frac{S-1}{\sqrt{V(S)}} & S > 0 \\ 0 & S = 0 \\ \frac{S+1}{\sqrt{V(S)}} & S < 0 \end{cases}.$$
(6)

The existence of autocorrelation in a dataset affects the probability of detecting a trend
when it does not exist and vice versa, but this is often ignored. Thus, the modified
nonparametric trend test developed by Hamed and Rao (1998) was applied in this
study. The corrected $Z$ value is derived as follows:
$$Z = \begin{cases} \frac{S-1}{\sqrt{V^*(S)}} & S > 0 \\ 0 & S = 0, \\ \frac{S+1}{\sqrt{V^*(S)}} & S < 0 \end{cases}$$
(7)

where
$$V^*(S) = V(S) * \frac{n}{n_S^*},$$
(8)





$$\frac{n}{n_S^*} = 1 + \frac{2}{n(n\text{-}1)(n-2)} * \sum_{i=1}^{n-1} (n\text{-}i)(n\text{-}i\text{-}1)(n\text{-}i-2)\rho_S(i)$$

$\qquad\qquad\qquad\qquad\qquad\qquad\qquad\qquad\qquad\qquad\qquad$ , (9)
where $\rho_S(i)$ is an autocorrelation function of the rank with respect to the observations.
The sign of $Z$ represents the trend direction and the magnitude of $Z$ is associated with
the significance level, where $|Z| > 1.64$ for the 10 % significance level and $|Z| > 1.96$ for
the 5 % significance level.

**2.4. Intentionally Biased Bootstrapping Method**
Bootstrapping analysis is a statistical method that can generate replicated datasets from
source data, and it can evaluate the variability of their quantiles without performing
separate analytical calculations (Davision et al., 2003). However, the intentionally
biased bootstrapping (IBB) technique applied in this study is a method that allows
assessment of the relative effects of a response variable by deliberately increasing or
decreasing the mean of the explanatory variable to a certain level while resampling it
with the response variable (Lee, 2017). A brief description of the IBB analysis process
is given below.
$\qquad$ Among $n$ observations $x_i$ ( $i = 1, 2, 3, \dots, n$ ), suppose that the mean of the
generated data is deliberately increased or decreased by $\Delta\mu$ for resampling of the
observations with bootstrapping. As a result, high (low) values are likely to be
resampled and low (high) values could be less likely to be selected. Thus, IBB can be
obtained by allocating different weights $S_{i,n}$ depending on the following observation
values (Eq. 10):
$\qquad\qquad\qquad\qquad\qquad\qquad S_{i,n} = i \, / \, n.$ $\qquad\qquad\qquad\qquad$ (10)
The weight $S_{i,n}$ assigned after scaling and adjustment contributes to the
probability of selection for the data observed in the IBB procedure. The average of the
resampled data can be expressed as in Eq. 11:
$$\tilde{\mu} = \frac{1}{\psi} \sum_{i=1}^{n} S_{i,n} x_i ,\qquad(11)$$

where $x_i$ represents the $i$-th incremental value and $\psi = \sum_{i=1}^{n} S_{i,n}$. The average
amount of increase or decrease $\Delta\mu$ is shown in Eq. 12:
$$\Delta\mu = \frac{1}{\psi} \sum_{i=1}^{n} S_{i,n} x_i - \frac{1}{n} \sum_{i=1}^{n} x_i.\qquad(12)$$

To obtain another value of $\Delta\mu$, the weights can be regeneralized in order of weight
sequence $(r)$; thus, $\Delta\tilde{\mu}(r)$ is derived as follows:
$$\Delta\tilde{\mu}(r) = \tilde{\mu}(r) - \hat{\mu} = \frac{1}{\psi_r} \sum_{i=1}^{n} s_{i,n}^{r} x_i - \frac{1}{n} \sum_{i=1}^{n} x_i.\qquad(13)$$

If the average value of increase or decrease is given as $\Delta\mu$, the weight "$r$" can be
calculated accordingly. In this study, the selection of the weight sequence was
performed using a Self-Organizing Migrating Algorithm (Zelinka, 2004) with the
objective function to minimize $[\Delta\mu - \Delta\tilde{\mu}(r)]^2$. In addition, the IBB technique was
employed to generate resampled datasets for the IOD and the response to the intensity
and frequency of rainfall to identify IOD-sensitive hotspots over the ICP. The statistical
significance of the analysis results was assessed using the significance level of the 95th
percentiles.

**3. Results**
**3.1. Seasonal Precipitation Patterns across the ICP**
The ICP is a region in which monsoon rains occur in different seasons in association
with seasonal winds and mountain areas. Geographically, the ICP has the Arakan



Mountains in the west, the Bilauktung Mountains and the Dawna Mountains in the
center, and the Annamese Mountains in the east. Meteorologically, the ICP is divided
into three monsoon periods: the southwest monsoon during June–November,
southeast monsoon during September–November, and northeast monsoon during
November–February. This study considered the wet season (May–October) and the
dry season (November–April) to identify the potential impact on regional rainfall
associated with atmosphere–ocean feedback in the Indian and Pacific oceans.
Figure 3 shows the seasonal average precipitation during the wet and dry seasons
across the ICP region during 1979–2018. The total precipitation during the wet season
across the ICP is about 1000–1500 mm. In addition, it has been confirmed that
precipitation variability is dependent on specific regions (Fig. 3a). The precipitation
variability was found to differ significantly between inland (<1000 mm) and coastal
areas (>2000 mm). Precipitation on the western coast of Cambodia, coast of western
Thailand, and Myanmar during June–November is attributable to the influence of the
southwest and southeast monsoons. Moreover, clear difference in precipitation is
evident between eastern and western parts of the Arakan Mountains in Myanmar. As
water vapor from Bangorman decreases over the mountains, the Arakan Mountains
show an arid climate to the east and a pattern of strong precipitation to the west.
During the dry season, total precipitation across the ICP is about 150–200 mm,
indicating that rainfall variability is not significantly dependent on specific regions
(Fig. 3b). In particular, in the dry season, because of the influence of the northeast
monsoon during November–February, high rainfall is received in central coastal areas
of Vietnam, e.g., near the city of Danang. Similarly, in the case of Myanmar, eastern
parts are dry because of the influence of the Arakan Mountains. The climatic
characteristics of the ICP are distinctive not only because of the effects of monsoons



and mountain areas, but also because of the characteristics of local areas and because of
specific temporal effects. The precipitation patterns of the ICP are likely to change
according to the characteristics of the wet and dry seasons, as well as because of the
influence of ocean-related climate factors (e.g., the IOD and ENSO).

**3.2. Spatiotemporal Variation in Precipitation over the ICP**
Figures 4 and 5 illustrate the long-term trend of precipitation over the ICP during 1979–
2018 for the wet and dry seasons, respectively. They show the results of the six major
climate change indices that represent the magnitude and frequency of precipitation. For
each figure, the direction of the trend is displayed in blue (increase) and red (decrease).
Figures 4a, 4b, 5a, and 5b show the long-term trends of PRCPTOT and R95pTOT.
These seasonal indices can be used to assess total precipitation. It can be seen that the
characteristics of their spatial distribution are similar. During the wet season, there is a
noticeable decrease in precipitation at the 5–10 % significance level in northern
Cambodia, some parts of Laos, and southern Thailand. In addition, it can be seen that
there is a marked trend of increase at the 5–10 % significance level in northwestern
Myanmar, parts of western Thailand, central Vietnam, and southern parts of China
(Fig. 4a and 4b).
During the dry season, there is a noticeable increase in precipitation at the 5–10 %
significance level along eastern and southern coastal areas of the ICP (i.e., Vietnam and
Cambodia) and some southern coastal regions of Thailand (Fig. 5a and 5b). The
R95pTOT climate index also shows a trend of increase in precipitation to the west of
the Arakan Mountains in Myanmar (Fig. 5b). Therefore, long-term changes in the
pattern of precipitation across the ICP during the wet season show a trend of decrease
(increase) in central inland areas (some coastal areas). During the dry season, there is a



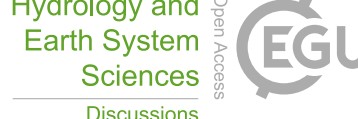

general trend of increase in precipitation across the ICP. Notably, the trend of increase
in precipitation in southeastern coastal areas appears significant.

Figures 4c, 4d, 5c, and 5d illustrate the long-term trends in RX1day and SDII. The

RX1day and SDII climate indices can be used to assess rainfall intensity. It can be seen
that the characteristics of the spatial distribution of the two indices are similar.
Moreover, the characteristics of their spatial distribution are also similar to PRCPTOT
and R95pTOT. It can be seen that during the rainy season the intensity of rainfall in
central and northern Myanmar, central and southern Vietnam, and southern China
increases, whereas the rainfall intensity decreases in Laos, Cambodia, northeastern
Myanmar, and South Vietnam. During the dry season, rainfall intensity generally
increases across the ICP, although it shows a clear pattern of decrease in Laos, as in the
wet season.

Figures 4e, 4f, 5e, and 5f show the long-term trends in CDD and CWD. The CDD

and CWD indices can be used in assessment of droughts and floods, respectively.
Therefore, it is unsurprising that the CDD and CWD indices exhibit opposite spatial
distribution characteristics. During the rainy season, the CDD value across the ICP
largely tends to increase, although it decreases in some coastal areas, e.g., Vietnam. The
CWD index shows the reverse tendency.

During the dry season, an increase (decrease) of the CDD (CWD) index can be

clearly observed at the 5–10 % significance level (Fig. 5e and 5f). The CDD index
increases along the southeast coast of the ICP, e.g., in areas of Vietnam, Cambodia, and
southern Thailand, whereas the CWD index exhibits the opposite trend. An increase
(decrease) in the CDD index suggests that drought is more (less) likely to occur, while a
decrease (increase) in the CWD index means that the occurrence of drought is less
(more) likely. Therefore, during the rainy season, floods are expected to increase along





the southeastern coast of the ICP (e.g., in Vietnam, Cambodia, and Thailand), while
drought is more likely to occur during the dry season.

**3.3. Precipitation Variability Associated with the IOD and ENSO**
The IOD, Asian monsoon, and other regional climatological patterns can lead to local
or global climate change, particularly in Indian Ocean Rim countries, which can cause
severe flooding or droughts depending on IOD variability (Lqbal and Hassan, 2018).
Composite analysis can clarify the role of the Southeast Asian Summer Monsoon in
precipitation variability across the ICP region associated with years of strong IOD and
ENSO, after identifying that tropical climate phenomena are the main factors that
influence precipitation variability over the ICP during the wet and dry seasons.
However, this role differs depending on the combination of the two climate
phenomena and on the season.

Figure 6 shows the results of composite rainfall anomalies (shown as a

percentage relative to normal) over the ICP during the wet and dry seasons in relation
to the IOD and ENSO. The patterns of rainfall anomalies indicate significant
difference between positive and negative IOD years. For positive IOD years, the wet
season rainfall (Fig. 6a) shows a decrease of <20 % in southern parts of the ICP,
whereas there is a marked increase in rainfall centered over the Arakan Mountains in
western Myanmar. It can be seen that the amount of rainfall received during the dry
season (Fig. 6c) is similar to that in the wet season, but there is 40–50 % less rainfall
than usual in certain mainland regions of Southeast Asia, especially Yangon and
Mawlamyine in Myanmar and in eastern Cambodia.

In negative IOD years, intense positive anomalies of rainfall can be seen in

central Cambodia and southern parts of Vietnam. A slight strong-pitched anomaly





pattern is evident during the wet season (Fig. 6b) around the coastline of both
Bangladesh and Myanmar, whereas weak-pitched positive anomalies (about 10–15 %
relative to the long-term average) are found throughout the ICP. However, changes in
rainfall pattern are not evident during the dry season (Fig. 6d), and although the
amount varies depending on region, rainfall is generally >30–50 % above the
long-term average. As in the wet season, the dry season also shows relatively strong
positive rainfall patterns with positive anomalies of >80–100 % in Cambodia and both
central and southern Vietnam.

Sometimes droughts and flooding are likely to converge because of remote

connections during IOD–ENSO periods, and they can have significant impact on the
modulation of the large-scale oceanic and atmospheric environment, especially in the
Indian Ocean and in Pacific Rim countries (Meza, 2013; Mahala et al., 2015; Lqbal
and Hassan, 2018). Thus, consideration of both combined and independent effects of
ENSO and the IOD on seasonal precipitation variability can provide improved
predictive expertise, and reveal new insight into tropical climate change and global
warming impacts (Ashok et al., 2001).

Figure 7 shows composite rainfall anomalies (November–April) during positive

and negative IOD years that coincided with ENSO. During positive IOD and El Niño
years (Fig. 7a), there is less rainfall than usual across Thailand, Cambodia, southern
Laos, and Vietnam. In particular, southern regions of Myanmar (from Yangon to
Mawlamyine) that border the Andaman Sea show a distinct decrease in rainfall by
more than 50 % in comparison with the long-term mean (1981–2010). However, in
contrast, there is 20–40 % more rainfall than usual in northern parts of the ICP, e.g.,
northern Myanmar, northeastern parts of Laos, and Vietnam. Furthermore, in
Guangzhou in China, rainfall is up to 60 % higher in comparison with average years.



These rainfall signals are stronger in WP El Niño years than in CT El Niño years
(figures not shown). During negative IOD and La Niña years (Fig. 7b), rainfall above
the long-term average is observed throughout the ICP, except for parts of central Laos
and northern Vietnam. The pattern of increased rainfall appears strongly throughout
Myanmar and regions around Ho Chi Minh City in Vietnam. However, in the region
adjacent to India and Bangladesh, as well as the Shenzhen area of China, strong
negative anomalies are evident.

**3.4. Identification of IOD-Sensitive Hotspots through IBB Simulations**
Section 3.3 discussed the significant impact on rainfall anomalies in the ICP
attributable to the combined or independent effects of ENSO and the IOD. In particular,
both positive IOD events and El Niño and negative IOD events and La Niña interact in
modulating rainfall anomalies over the ICP. The IOD and ENSO are strongly correlated
and their variations are mutually forced or triggered (Yu and Lau, 2005; Yuan and Li,
2008; Lestari and Koh, 2016). For the period 1979–2017, the correlation between the
peak phase of the IOD and the two types of El Niño index proposed by Ren and Jin
(2011) was analyzed. The results showed the IOD has strong positive correlation with
the CT El Niño ($N_{CT}$) ($\rho = 0.4850$, p-value = 0.0018). However, the IOD also has
positive correlation with the WP El Niño ($N_{WP}$), but not at a statistically significant
level ($\rho = 0.110$, p-value = 0.5013). These results are also reflected in the results of
the IBB simulation (Fig. 8). Figure 8 shows the results of 1000 simulations for the
$N_{CT}$ and $N_{WP}$ indices performed by applying the IBB technique to the IOD index
based on historical observations for the period 1979–2017. For applying a +1SD
increase of the IOD, the mean difference between the observation of $N_{CT}$ and
simulated $N_{CT}$ shows a statistically significant increase at the 95 % significance level



(diff. = 0.392, Interquartile range (IQR) = 0.228). However, the difference in the
mean value of the $N_{WP}$ index, although increased slightly, is not statistically
significant (diff. = 0.097, IQR = 0.094). For applying a −1SD decrease of the IOD, the
simulation results show changes similar to the case with a +1SD increase of the IOD
($N_{CT}$: diff. = 0.360, IQR = 0.108, $N_{WP}$: diff. = 0.088, IQR = 0.098). Therefore, for
changes in the IOD, the linear increase (or decrease) in the $N_{CT}$ index is more
pronounced than the change in the $N_{WP}$ index.

The spatiotemporal connection between SST and winds shows strong coupling

through precipitation and ocean dynamics (Saji et al., 1999). This dipole mode,
accounts for about 12 % of SST variability in the Indian Ocean, and its duration of
activity can greatly affect both the intensity and the frequency of rainfall in the Indian
Ocean Rim countries, including the ICP. Based on statistical simulations of historical
observations (1979–2018), Figs. 9 and 10 show rainfall variation and the most
sensitive hotspot areas in the wet and dry seasons of the ICP attributable to IOD
changes.

The spatial distribution of differences in PRCPTOT is shown in Fig. 9, given the

condition of a ±1SD increase or decrease of the IOD in the wet season. For a +1SD
increase of the IOD, PRCPTOT is >90 % higher than usual throughout Myanmar, and
weak positive anomaly patterns are evident in southwestern China. In contrast, a
pattern of decrease of PRCPTOT of 15–20 % less than the long-term average is evident
in Cambodia and southern Vietnam, i.e., in areas of the downstream reaches of the
Mekong River. However, no statistically significant changes occur in the central ICP
region, except in some parts of central Laos and Thailand. This spatial distribution of
rainfall anomaly is also found for the RX1day index, although occasional patterns of
increase or decrease are evident and the spatial extent is reduced. In addition,



throughout Myanmar, the CDD index is decreased by >25 % in comparison with the
long-term average year, while the CWD index is increased by 35–50 %. For the CDD
index, a statistically significant pattern of decrease is found across Vietnam, Cambodia,
and Laos. The most significant changes in the CWD index are across Myanmar
(increase of 35–50 %), southern Cambodia, and the southeast coast of Vietnam
(decrease of 15–20 %). The other ICP regions generally show a pattern of weak
increase in terms of CWD. For a −1SD decrease of the IOD, PRCPTOT, RX1day, and
CWD all show distinct patterns of increase in the Laos and Vietnam basins, while the
CDD index shows a predominant pattern of decrease, except in certain areas. Analysis
indicates that other regions have a reverse pattern compared with the case of the +1SD
increase of the IOD. Consequently, it is determined that changes in rainfall during the
wet season in the ICP region are sensitive to changes in the IOD.

Given the condition of a ±1SD increase or decrease of the IOD for the dry season,

the spatial distribution of the rainfall indices is shown in Fig. 10. For a decrease of
−1SD of the IOD, there is more rainfall (PRCPTOT and RX1day) than usual
throughout the ICP, especially in Laos and Vietnam. For a +1SD increase of the IOD,
negative anomaly patterns of PRCPTOT are dominant in southern Vietnam, eastern
Cambodia, and northeastern Thailand, while weak patterns of positive anomaly are
evident in areas of Myanmar and South China. Compared with the changes in the
rainfall indices during the wet season, changes in the rainfall indices are intensified
and the spatial influence is more extensive. However, for the CDD and CWD indices,
either the positive anomaly patterns are weakened or negative anomaly patterns
appear for a +1SD increase of the IOD. Especially for the CWD index, a pattern of
decrease by more than 10–20 % compared with the long-term average is found in
Thailand, whereas the Myanmar region shows a pattern of increase of 15–25 %. In

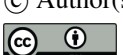



this study, we simulated the changes in both wet and dry season rainfall across the
ICP according to intentional IOD changes, and IOD-sensitive hotspots were verified
through quantitative analysis. The findings of this study could help elucidate the
long-term changes in rainfall expected in the ICP region in a changing climate.

**4. Summary and Conclusions**
This study analyzed changes in the magnitude and frequency of precipitation during the
dry and wet seasons over the ICP, taking into account both the dipole mode in the
tropical Indian Ocean and SST warming in the Pacific Ocean. The main results are
summarized in the following.
1.  According to analyses of the long-term trend in seasonal rainfall across the ICP
during 1979–2018, rainfall showed significant decreases in northern Cambodia,
parts of Laos, and southern Thailand during the wet season (May–October).
Moreover, significant increases were evident in northwestern Myanmar, some
parts of western Thailand, central Vietnam, and southern China. During the dry
season (November–April), PRCPTOT rose noticeably in eastern and southern
coastal areas of the ICP (i.e., Vietnam and Cambodia) and some southern
coastal regions of Thailand.
2.  During the wet season, the CDD index increased and decreased in some coastal
areas such as Vietnam. However, during the dry season, increases in CDD and
decreases in CWD were evident in the ICP. In particular, a pattern of decline in
CWD dominated southeastern coastal areas of the ICP, including Vietnam,
Cambodia, and southern Thailand.
3.  The IOD showed strong positive correlation with the CT El Niño. However,
although the IOD exhibited positive correlation with the WP El Niño, the



relationship was not statistically significant. The variability of rainfall across
the ICP was confirmed amplified by combined and independent effects of
ENSO and the IOD. During years of positive IOD and El Niño, there was less
rainfall than usual throughout Thailand and Cambodia, southern Laos, and
Vietnam. In particular, the southern part of Myanmar, which borders the
Andaman Sea, showed a decrease in regional rainfall of >50 % in comparison
with the long-term average. In contrast, northern parts of India and China,
including Myanmar, northeastern Laos, and Vietnam, received 20–40 % more
rainfall than usual. Years with a negative IOD mode and La Niña showed
rainfall above the long-term average across the ICP, except for certain parts,
e.g., Central Laos and northern Vietnam.
4. Through application of the IBB technique, this study simulated the change of
rainfall across the ICP for the wet and dry seasons according to intentional IOD
changes, and IOD-sensitive hotspots were verified through quantitative analysis.
For the wet season, a +1SD increase of the IOD resulted in >90 % more
PRCPTOT than usual across Myanmar in the northwestern ICP. Conversely, in
Cambodia and southern Vietnam, rainfall patterns were 15–20 % less than the
long-term average in the region of the lower Mekong River. In addition, the
CDD index decreased throughout Myanmar by >25 % compared with the
long-term average. The most significant change in the CWD index was in
Myanmar, i.e., a 35–50 % increase. However, a pattern of decrease appeared
across the southeastern coast of the ICP in southern Cambodia and Vietnam.
For a +1SD increase of the IOD in the dry season, negative anomaly patterns of
PRCPTOT were found dominant in South Vietnam, eastern Cambodia, and
northeastern Thailand, and more rainfall than usual occurred throughout the





ICP, especially in Laos and Vietnam, when considering a −1SD decrease of the
IOD.

Although the results of this study are based on limited observations, they provide a
clear perspective on the sensitivity of local precipitation to atmosphere–ocean
interactions, and they reveal the potential future impact of statistical changes to the IOD,
improving our understanding of the associated regional impact on precipitation under
the effects of climate change.
**Author contribution:** conceptualization, J.K., S.Y., and T.L.; Formal analysis, J.K.;
Methodology, T.L. and J.K.; resources, J.K. and L.X.; writing—original draft
preparation, P.X., S.Y., and J.K.; writing—review and editing, L.X. and T.L.
**Acknowledgments:** This research is supported by the National Key R&D Program of
China (2017YFC0405901), the National Natural Science Foundation of China (No.
51525902), and the Ministry of Education "111 Project" Fund of China (B18037), all
of which are greatly appreciated.
**Competing interests:** The authors declare that they have no conflict of interest.

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



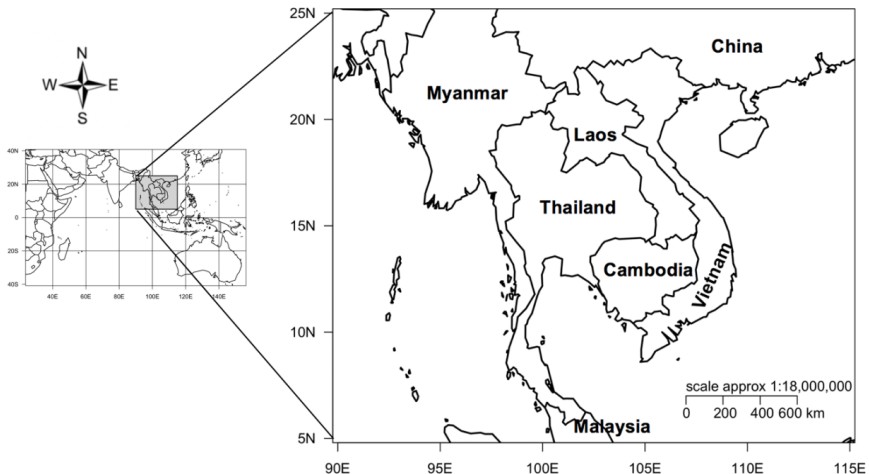


**Figure 1.** Map of the Indochina Peninsula (5°–25° N, 90°–115° E).



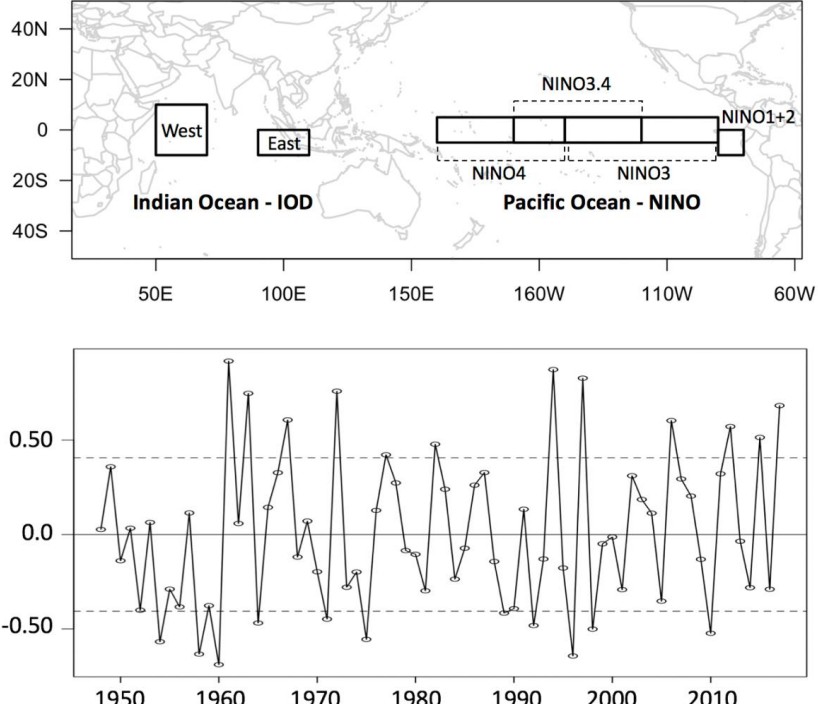


**Figure 2.** Dipole mode in the tropical Indian Ocean (TIO) and Niño region in the Pacific
Ocean. The Indian Ocean Dipole (IOD) index is defined based on the sea surface temperature
anomaly difference between the western (10° S–10° N, 50°–70° E) and southeastern (10° S to
the equator, 90°–110° E) regions of the TIO shown in the upper panel. In the lower panel, the
IOD time series during 1948–2017 is shown by the solid line, and the ±1SD of the IOD is
marked by dotted lines.





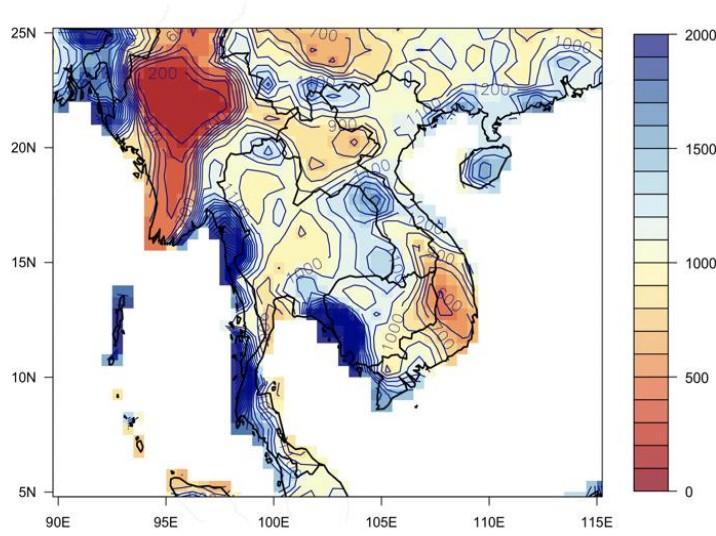

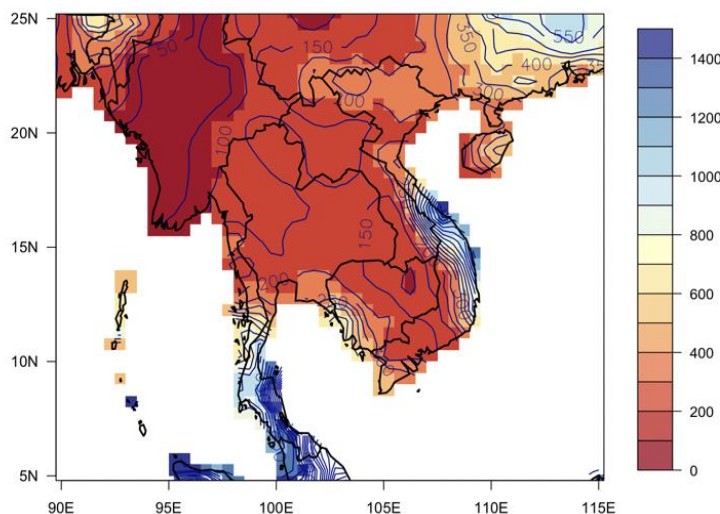


**Figure 3.** Average precipitation (mm) during the (a) wet and (b) dry seasons (1979–2018).

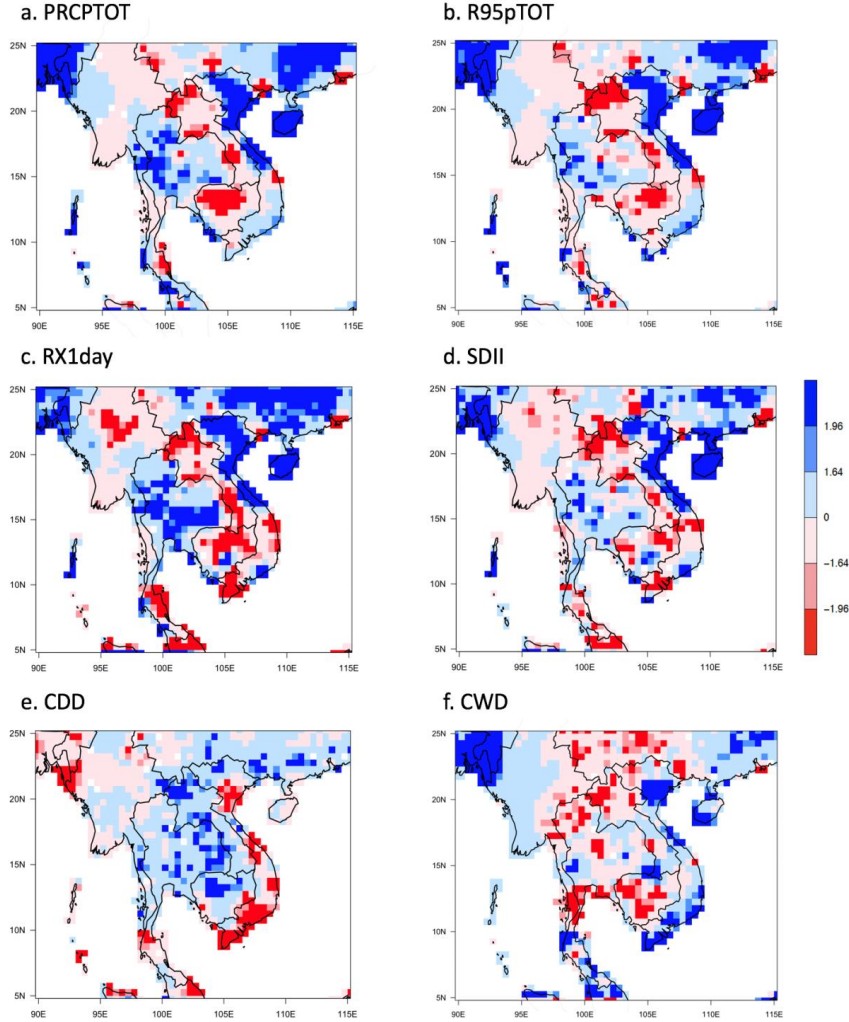


**Figure 4.** Long-term trend in seasonal precipitation for the wet season (May–October) over
the ICP during 1979–2018. (a)–(f) show the analysis results of the six major climate change
indices that reflect the magnitude and frequency of precipitation. In each panel, positive and
negative trends are displayed in blue and red, respectively. The magnitude of Z is associated
with the significance level, i.e., $|Z| > 1.64$ is for the 10 % significance level and $|Z| > 1.96$ is
for the 5 % significance level.


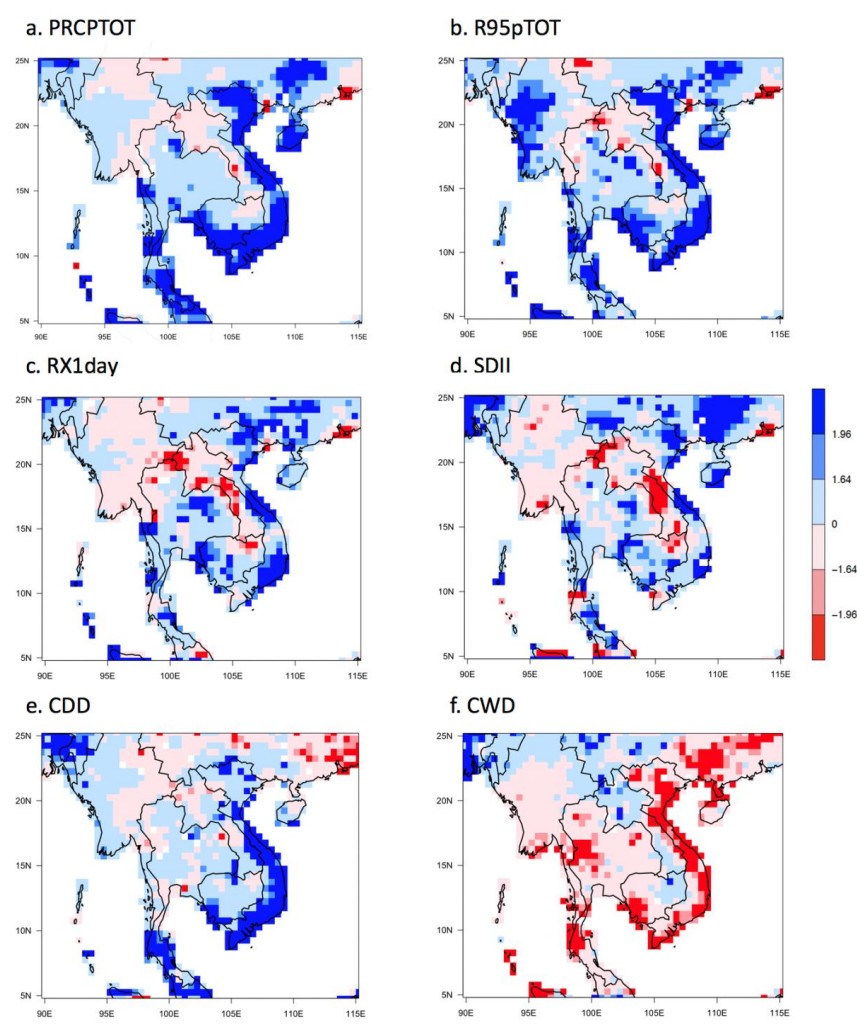

**Figure 5.** Same as Fig. 4 but for seasonal precipitation during the dry season (November–
April).

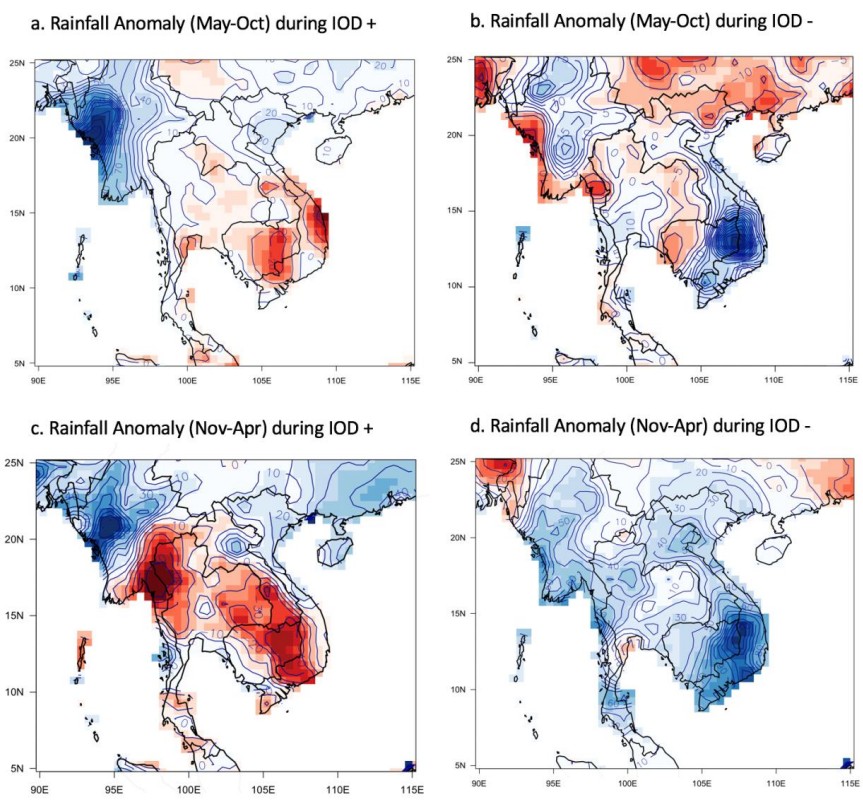


**Figure 6.** Composite of seasonal rainfall anomaly (%) during positive and negative IOD years: (a) rainfall anomaly in wet season during positive IOD years, (b) rainfall anomaly in wet season during negative IOD years, (c) rainfall anomaly in dry season during positive IOD years, and (d) rainfall anomaly in dry season during negative IOD years. Positive (negative) values show increasing (decreasing) rainfall departure from the long-term average (1981–2010).

651

### a. Rainfall Anomaly during IOD + & El Nino

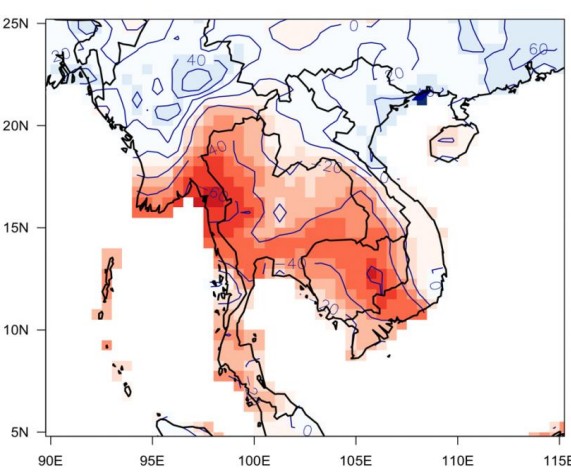

### b. Rainfall Anomaly during IOD - & La Nina

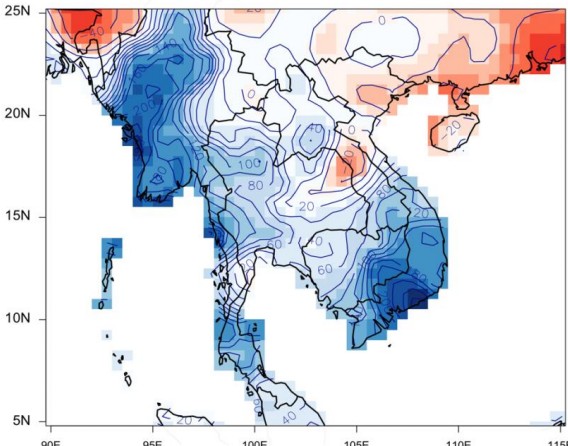

652

**Figure 7.** Composite rainfall anomaly in dry season (November–April) associated with the
IOD and ENSO: (a) rainfall anomaly during years with positive IOD and El Niño, and (b)
rainfall anomaly during years with negative IOD and La Niña. Positive (negative) values
show increasing (decreasing) rainfall departure from the long-term average (1981–2010).





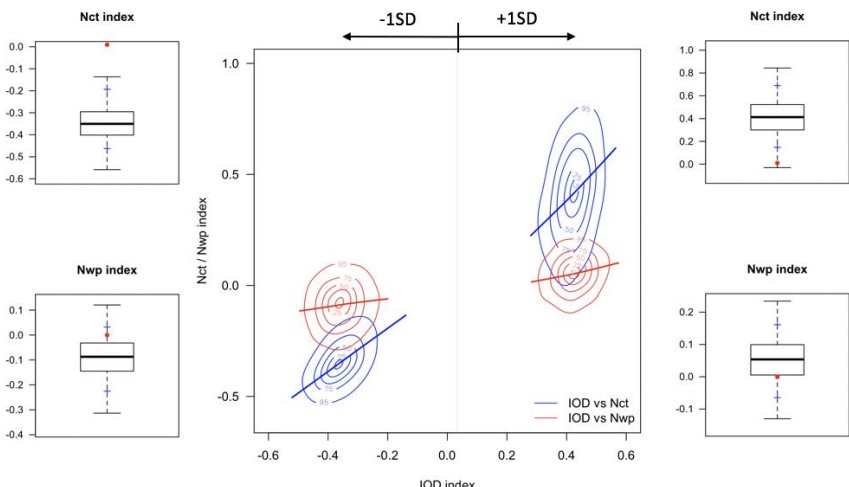

657

**Figure 8.** Mean differences of the two types of El Niño with ±1SD of the IOD. In the main
panel, contours (5th, lower quadrant, median, upper volatile, and 95th level) summarize the
IOD index and Nct or Nwp index using the intentionally biased bootstrapping model. Both
left and right panels deliberately apply ±1SD of the IOD to show results of 1000 simulations
for the Nct and Nwp indices. Red dots in each panel represent the average value of the
observations.

664

665

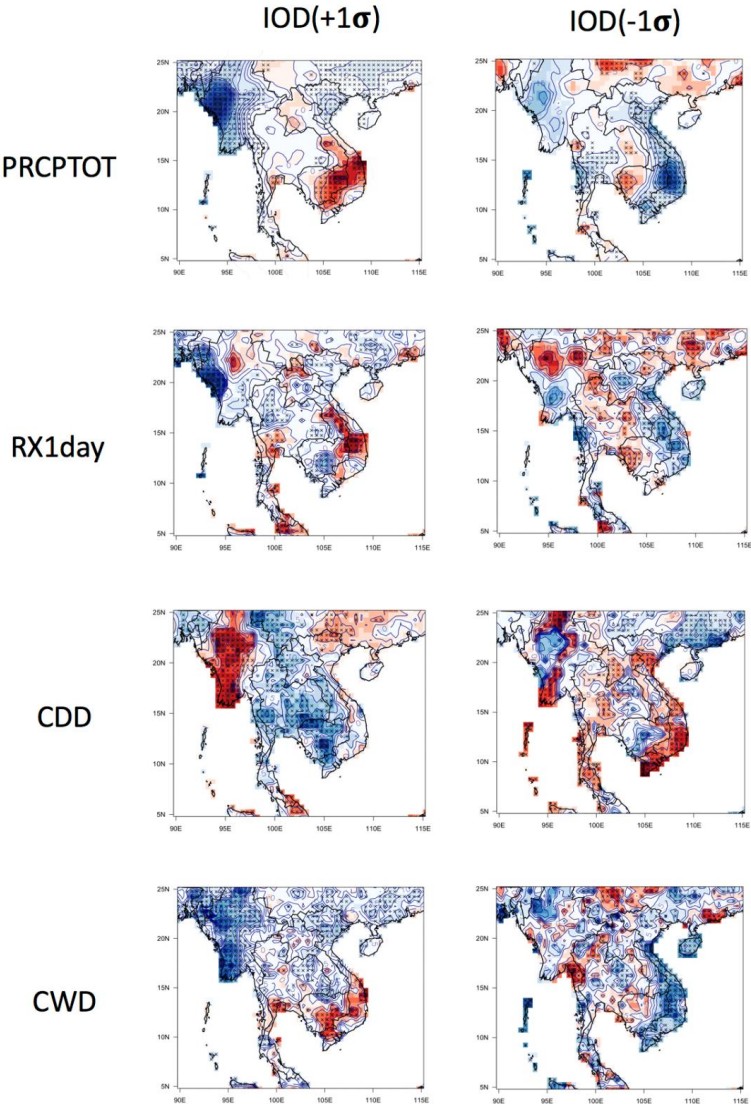

666
**Figure 9.** Spatial distributions of the percentage changes in major precipitation indices for the
wet season (May–October) over the ICP region for intentional increases (+1SD) or decreases
(−1SD) of the IOD index using the intentionally biased bootstrapping simulation. In each
panel, the statistically significant area of change at the 95 % significance level is shown by an
"x" symbol.



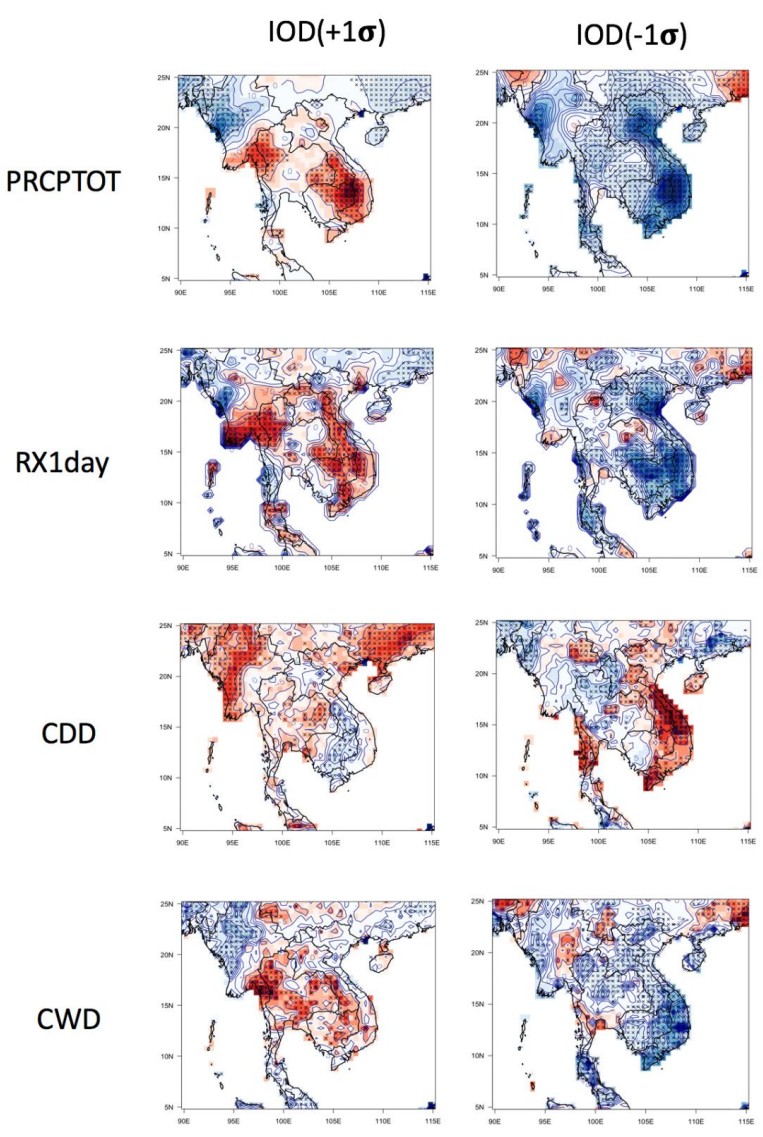

672

**Figure 10.** Same as Fig. 9 but for the dry season (November–April) over the ICP region.

674