# Peer review of "Identification of Hotspots of Rainfall Variation Sensitive to Indian Ocean Dipole Mode through Intentional Statistical Simulations"

_Hydrology and Earth System Sciences, 2019_

## Referee Comment (RC1) · Anonymous Referee #1 · 1 Jul 2019

This study analyzed changes in the magnitude and frequency of precipitation during the dry and wet seasons over the Indochina Peninsula, taking into account both the dipole mode in the tropical Indian Ocean and SST warming in the Pacific Ocean. Although the topic seems interesting, this study needs substantial improvement in the presentation and physical mechanisms underlying the association between IOD and/or ENSO and precipitation in the Indochina Peninsula. Overall, this study does not meet the standard of this journal and I am reluctant to recommend this study for publication in HESS. Below are some specific comments. Major comments: 1. When the authors discuss the association between IOD/ENSO on during the dry and wet seasons over the Indochina Peninsula, there is no interpretation for this association. For example, how

could the IOD, along with ENSO impact precipitation during different seasons, through what mechanisms? How do IOD and ENSO influence moisture transport and convergence in this region? This is very basic understanding of such association. However, I couldn't find any related analysis along this line. 2. The impacts of different types of ENSO on precipitation in the Indochina Peninsula during different seasons and phases of ENSO have been reported extensively in climate science. From this perspective, it is not a novel angle to examine this question. More in-depth analyses are required to advance our understanding. 3. The title of this manuscript only includes "IOD", but this study focuses on IOD and ENSO. This is really confusing.

Minor comments: Line 56: the impacts of these new ... Lines 56-60: The authors may rewrite this sentence to clarify what they intend to express. Lines 65-68: Which water than normal? Is it ocean surface or subsurface water? Commonly, it is described as sea surface temperature anomaly, rather than water warmer than normal. Lines 78-80: The hiatus of global warming is tied to the considerable transport of heat from the Pacific Ocean into the Indian Ocean via the Indonesian Throughflow (Lee et al. 2015), while Kosaka and Xie (2013) found that the La Nino-like sea surface temperature change was responsible for the hiatus. Please carefully cite references. Lines 81 and 84: Zhang et al. (2018) was cited here, but not included in the reference list. Figures 6, 7, 9 and 10: There is no color bar for these figures. This is unacceptable because a color figure without a color bar means nothing.

---

## Referee Comment (RC2) · Anonymous Referee #2 · 31 Jul 2019

In this study authors tried to identify the regions of Indochina peninsula affected by IOD. There is a lack of clarity in the presentation and several issues need to be addressed. i) It is not clear what the authors mean by +1sd and -1sd IOD. Is it the IOD index created for the whole season, dry and wet separate or have the authors identified it from Fig 2. ii) If authors are creating composites of dry and wet season, then the IOD index should be based on seasonal averages iii) The years that have been used in the composites are to be mentioned in the text. iv) Authors need to carry out significance test for the composites. v) Lines 84-87: Authors needs to review or cite papers which have already carried out the studies relating to IOD and ENSO over the region. For eg. Tsai et al. (2015) Indo-China monsoon indices, Scientific reports. vi) Piechota

et al have not discussed about IOD in their paper, so reference to the paper should be removed from some lines. vii) Lines 324-339: As indicated in the introduction IOD initiates sometime in May or June and ends by November. So, the affect of IOD on the dry season is negligible and analyzing the affect of IOD on that season is not useful.

---

## Editor Comment (EC1) · Louise Slater (Editor) · 19 Aug 2019

Dear Authors,

We have received two reviews of your manuscript, "Identification of Hotspots of Rainfall Variation Sensitive to Indian Ocean Dipole Mode through Intentional Statistical Simulations". After a careful re-reading of the manuscript as well as the two reviews, I am sorry to say that I cannot recommend publication of your work in HESS, and therefore I recommend withdrawal of the manuscript (in this case, the response to reviewers is not required).

[Figure]

The reviewers both raise some valid concerns regarding the work, namely regarding: the clarity of the methods (description of the indices, their construction, and the discussion of the physical mechanisms involved); the novelty of the study (including how it is framed in light of earlier research); the attention to detail (including the design of the figures, and utility of the dry season analysis).

I hope these suggestions may be helpful in revising the manuscript and resubmitting it elsewhere.

Sincerely, Louise Slater